# Neuro-Symbolic Program Synthesis

**Emilio Parisotto**[1,2]**, Abdel-rahman Mohamed**[1]**, Rishabh Singh**[1]**,**
**Lihong Li**[1]**, Dengyong Zhou**[1]**, Pushmeet Kohli**[1]
[1]Microsoft Research, USA    [2]Carnegie Mellon University, USA
eparisot@andrew.cmu.edu , {asamir,risin,lihongli,denzho,pkohli}@microsoft.com

## Abstract

Recent years have seen the proposal of a number of neural architectures for the problem of Program Induction. Given a set of input-output examples, these architectures are able to learn mappings that generalize to new test inputs. While achieving impressive results, these approaches have a number of important limitations: (a) they are computationally expensive and hard to train, (b) a model has to be trained for each task (program) separately, and (c) it is hard to interpret or verify the correctness of the learnt mapping (as it is defined by a neural network). In this paper, we propose a novel technique, *Neuro-Symbolic Program Synthesis*, to overcome the above-mentioned problems. Once trained, our approach can automatically construct computer programs in a domain-specific language that are consistent with a set of input-output examples provided at test time. Our method is based on two novel neural modules. The first module, called the cross correlation I/O network, given a set of input-output examples, produces a continuous representation of the set of I/O examples. The second module, the Recursive-Reverse-Recursive Neural Network (R3NN), given the continuous representation of the examples, synthesizes a program by incrementally expanding partial programs. We demonstrate the effectiveness of our approach by applying it to the rich and complex domain of regular expression based string transformations. Experiments show that the R3NN model is not only able to construct programs from new input-output examples, but it is also able to construct new programs for tasks that it had never observed before during training.

## 1 Introduction

The act of programming, *i.e.*, developing a procedure to accomplish a task, is a remarkable demonstration of the reasoning abilities of the human mind. Expectedly, *Program Induction* is considered as one of the fundamental problems in Machine Learning and Artificial Intelligence. Recent progress on deep learning has led to the proposal of a number of promising neural architectures for this problem. Many of these models are inspired from computation modules (CPU, RAM, GPU) (Graves et al., 2014; Kurach et al., 2015; Reed & de Freitas, 2015; Neelakantan et al., 2015) or common data structures used in many algorithms (stack) (Joulin & Mikolov, 2015). A common thread in this line of work is to specify the atomic operations of the network in some differentiable form, allowing efficient end-to-end training of a neural controller, or to use reinforcement learning to make hard choices about which operation to perform. While these results are impressive, these approaches have a number of important limitations: (a) they are computationally expensive and hard to train, (b) a model has to be trained for each task (program) separately, and (c) it is hard to interpret or verify the correctness of the learnt mapping (as it is defined by a neural network). While some recently proposed methods (Kurach et al., 2015; Gaunt et al., 2016; Riedel et al., 2016; Bunel et al., 2016) do learn interpretable programs, they still need to learn a separate neural network model for each individual task.

Motivated by the need for model interpretability and scalability to multiple tasks, we address the problem of *Program Synthesis*. Program Synthesis, the problem of automatically constructing programs that are consistent with a given specification, has long been a subject of research in Computer Science (Biermann, 1978; Summers, 1977). This interest has been reinvigorated in recent years on

the back of the development of methods for learning programs in various domains, ranging from low-level bit manipulation code (Solar-Lezama et al., 2005) to data structure manipulations (Singh & Solar-Lezama, 2011) and regular expression based string transformations (Gulwani, 2011).

Most of the recently proposed methods for program synthesis operate by searching the space of programs in a Domain-Specific Language (DSL) instead of arbitrary Turing-complete languages. This hypothesis space of possible programs is huge (potentially infinite) and searching over it is a challenging problem. Several search techniques including enumerative (Udupa et al., 2013), stochastic (Schkufza et al., 2013), constraint-based (Solar-Lezama, 2008), and version-space algebra based algorithms (Gulwani et al., 2012) have been developed to search over the space of programs in the DSL, which support different kinds of specifications (examples, partial programs, natural language etc.) and domains. These techniques not only require significant engineering and research effort to develop carefully-designed heuristics for efficient search, but also have limited applicability and can only synthesize programs of limited sizes and types.

In this paper, we present a novel technique called *Neuro-Symbolic Program Synthesis (NSPS)* that learns to generate a program incrementally without the need for an explicit search. Once trained, NSPS can automatically construct computer programs that are consistent with any set of input-output examples provided at test time. Our method is based on two novel module neural architectures. The first module, called the cross correlation I/O network, produces a continuous representation of any given set of input-output examples. The second module, the Recursive-Reverse-Recursive Neural Network (R3NN), given the continuous representation of the input-output examples, synthesizes a program by incrementally expanding partial programs. R3NN employs a tree-based neural architecture that sequentially constructs a parse tree by selecting which non-terminal symbol to expand using rules from a context-free grammar (*i.e.*, the DSL).

We demonstrate the efficacy of our method by applying it to the rich and complex domain of regular-expression-based syntactic string transformations, using a DSL based on the one used by Flash-Fill (Gulwani, 2011; Gulwani et al., 2012), a Programming-By-Example (PBE) system in Microsoft Excel 2013. Given a few input-output examples of strings, the task is to synthesize a program built on regular expressions to perform the desired string transformation. An example task that can be expressed in this DSL is shown in Figure 1, which also shows the DSL.

Our evaluation shows that NSPS is not only able to construct programs for known tasks from new input-output examples, but it is also able to construct completely new programs that it had not observed during training. Specifically, the proposed system is able to synthesize string transformation programs for 63% of tasks that it had not observed at training time, and for 94% of tasks when 100 program samples are taken from the model. Moreover, our system is able to learn 38% of 238 real-world FlashFill benchmarks.

To summarize, the key contributions of our work are:

- A novel Neuro-Symbolic program synthesis technique to encode neural search over the space of programs defined using a Domain-Specific Language (DSL).

- The R3NN model that encodes and expands partial programs in the DSL, where each node has a global representation of the program tree.

- A novel cross-correlation based neural architecture for learning continuous representation of sets of input-output examples.

- Evaluation of the NSPS approach on the complex domain of regular expression based string transformations.

## 2   PROBLEM DEFINITION

In this section, we formally define the DSL-based program synthesis problem that we consider in this paper. Given a DSL $L$, we want to automatically construct a synthesis algorithm $\mathcal{A}$ such that given a set of input-output example, $\{(i_1, o_1), \cdots, (i_n, o_n)\}$, $\mathcal{A}$ returns a program $P \in L$ that conforms to the input-output examples, *i.e.*,

$$\forall j : 1 \leq j \leq n \ P(i_j) = o_j. \tag{1}$$

| | Input $v$ | Output |
|---|---|---|
| 1 | William Henry Charles | Charles, W. |
| 2 | Michael Johnson | Johnson, M. |
| 3 | Barack Rogers | Rogers, B. |
| 4 | Martha D. Saunders | Saunders, M. |
| 5 | Peter T Gates | Gates, P. |

(a)

$$
\begin{aligned}
\text{String } e \quad &:= \quad \text{Concat}(f_1, \cdots, f_n) \\
\text{Substring } f \quad &:= \quad \text{ConstStr}(s) \\
&\quad | \quad \text{SubStr}(v, p_l, p_r) \\
\text{Position } p \quad &:= \quad (r, k, \text{Dir}) \\
&\quad | \quad \text{ConstPos}(k) \\
\text{Direction Dir} \quad &:= \quad \text{Start} \mid \text{End} \\
\text{Regex } r \quad &:= \quad s \mid T_1 \cdots \mid T_n
\end{aligned}
$$

(b)

Figure 1: An example FlashFill task for transforming names to lastname with initials of first name, and (b) The DSL for regular expression based string transformations.

The syntax and semantics of the DSL for string transformations is shown in Figure 1(b) and Figure 8 respectively. The DSL corresponds to a large subset of FlashFill DSL (except conditionals), and allows for a richer class of substring operations than FlashFill. A DSL program takes as input a string $v$ and returns an output string $o$. The top-level string expression $e$ is a concatenation of a finite list of substring expressions $f_1, \cdots, f_n$. A substring expression $f$ can either be a constant string $s$ or a substring expression, which is defined using two position logics $p_l$ (left) and $p_r$ (right). A position logic corresponds to a symbolic expression that evaluates to an index in the string. A position logic $p$ can either be a constant position $k$ or a token match expression $(r, k, \text{Dir})$, which denotes the $\text{Start}$ or $\text{End}$ of the $k^{\text{th}}$ match of token $r$ in input string $v$. A regex token can either be a constant string $s$ or one of 8 regular expression tokens: $p$ (ProperCase), $C$ (CAPS), $l$ (lowercase), $d$ (Digits), $\alpha$ (Alphabets), $\alpha n$ (Alphanumeric), $^\wedge$ (StartOfString), and $ (EndOfString). The semantics of the DSL programs is described in the appendix.

A DSL program for the name transformation task shown in Figure 1(a) that is consistent with the examples is: $\text{Concat}(f_1, \text{ConstStr}(\text{", "}), f_2, \text{ConstStr}(\text{"."}))$, where $f_1 \equiv \text{SubStr}(v, (\text{" "}, -1, \text{End}), \text{ConstPos}(-1))$ and $f_2 \equiv \text{SubStr}(v, \text{ConstPos}(0), \text{ConstPos}(1))$. The program concatenates the following 4 strings: i) substring between the end of last whitespace and end of string, ii) constant string ", ", iii) first character of input string, and iv) constant string ".".

## 3  OVERVIEW OF OUR APPROACH

We now present an overview of our approach. Given a DSL $L$, we learn a generative model of programs in the DSL $L$ that is conditioned on input-output examples to efficiently search for consistent programs. The workflow of our system is shown in Figure 2, which is trained end-to-end using a large training set of programs in the DSL together with their corresponding input-output examples. To generate a large training set, we uniformly sample programs from the DSL and then use a *rule-based strategy* to compute well-formed input strings. Given a program P (sampled from the DSL), the rule-based strategy generates input strings for the program P ensuring that the pre-conditions of P are met (i.e. P doesn't throw an exception on the input strings). It collects the pre-conditions of all Substring expressions present in the sampled program P and then generates inputs conforming to them. For example, let's assume the sampled program is $\text{SubStr}(v, (\text{CAPS}, 2, \text{Start}), (\text{" "}, 3, \text{Start}))$, which extracts the substring between the start of $2^{\text{nd}}$ capital letter and start of $3^{\text{rd}}$ whitespace. The rule-based strategy would ensure that all the generated input strings consist of at least 2 capital letters and 3 whitespaces in addition to other randomly generated characters. The corresponding output strings are obtained by running the programs on the input strings.

A DSL can be considered as a context-free grammar with a start symbol $S$ and a set of non-terminals with corresponding expansion rules. The (partial) grammar derivations or trees correspond to (partial) programs. A naïve way to perform a search over the programs in a DSL is to start from the start symbol $S$ and then randomly choose non-terminals to expand with randomly chosen expansion rules until reaching a derivation with only terminals. We, instead, learn a generative model over partial derivations in the DSL that assigns probabilities to different non-terminals in a partial derivation and corresponding expansions to guide the search for complete derivations.

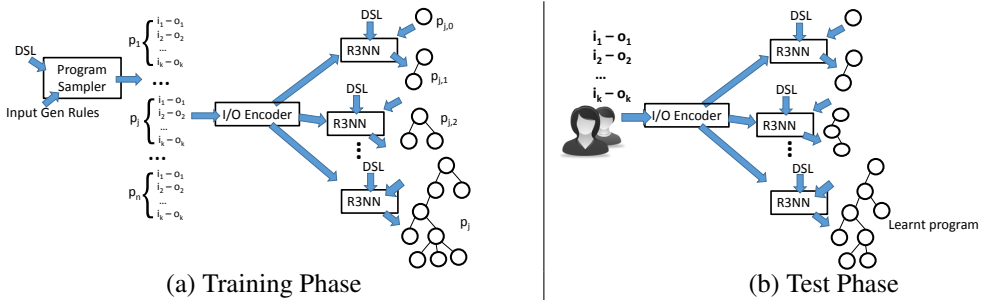

(a) Training Phase (b) Test Phase

Figure 2: An overview of the training and test workflow of our synthesis appraoch.

Our generative model uses a Recursive-Reverse-Recursive Neural Network (R3NN) to encode partial trees (derivations) in $L$, where each node in the partial tree encodes global information about every other node in the tree. The model assigns a vector representation for every symbol and every expansion rule in the grammar. Given a partial tree, the model first assigns a vector representation to each leaf node, and then performs a recursive pass going up in the tree to assign a global tree representation to the root. It then performs a reverse-recursive pass starting from the root to assign a global tree representation to each node in the tree.

The generative process is conditioned on a set of input-output examples to learn a program that is consistent with this set of examples. We experiment with multiple input-output encoders including an LSTM encoder that concatenates the hidden vectors of two deep bidirectional LSTM networks for input and output strings in the examples, and a Cross Correlation encoder that computes the cross correlation between the LSTM tensor representations of input and output strings in the examples. This vector is then used as an additional input in the R3NN model to condition the generative model.

## 4 TREE-STRUCTURED GENERATION MODEL

We define a program t-steps into construction as a partial program tree (PPT) (see Figure 3 for a visual depiction). A PPT has two types of nodes: leaf (symbol) nodes and inner non-leaf (rule) nodes. A leaf node represents a symbol, whether non-terminal or terminal. An inner non-leaf node represents a particular production rule of the DSL, where the number of children of the non-leaf node is equivalent to the arity of the RHS of the rule it represents. A PPT is called a program tree (PT) whenever all the leaves of the tree are terminal symbols. Such a tree represents a completed program under the DSL and can be executed. We define an expansion as the valid application of a specific production rule (e → e op2 e) to a specific non-terminal leaf node within a PPT (leaf with symbol e). We refer to the specific production rule that an expansion is derived from as the expansion type. It can be seen that if there exist two leaf nodes ($l_1$ and $l_2$) with the same symbol then for every expansion specific to $l_1$ there exists an expansion specific to $l_2$ with the same type.

### 4.1 RECURSIVE-REVERSE-RECURSIVE NEURAL NETWORK

In order to define a generation model over PPTs, we need an efficient way of assigning probabilities to every valid expansion in the current PPT. A valid expansion has two components: first the production rule used, and second the position of the expanded leaf node relative to every other node in the tree. To account for the first component, a separate distributed representation for each production rule is maintained. The second component is handled using an architecture where the forward propagation resembles belief propagation on trees, allowing a notion of global tree state at every node within the tree. A given expansion probability is then calculated as being proportional to the inner product between the production rule representation and the global-tree representation of the leaf-level non-terminal node. We now describe the design of this architecture in more detail.

The R3NN has the following parameters for the grammar described by a DSL (see Figure 3):

1. For every symbol $s \in S$, an $M-$dimensional representation $\phi(s) \in \mathbb{R}^M$.
2. For every production rule $r \in R$, an $M-$dimensional representation $\omega(r) \in \mathbb{R}^M$.

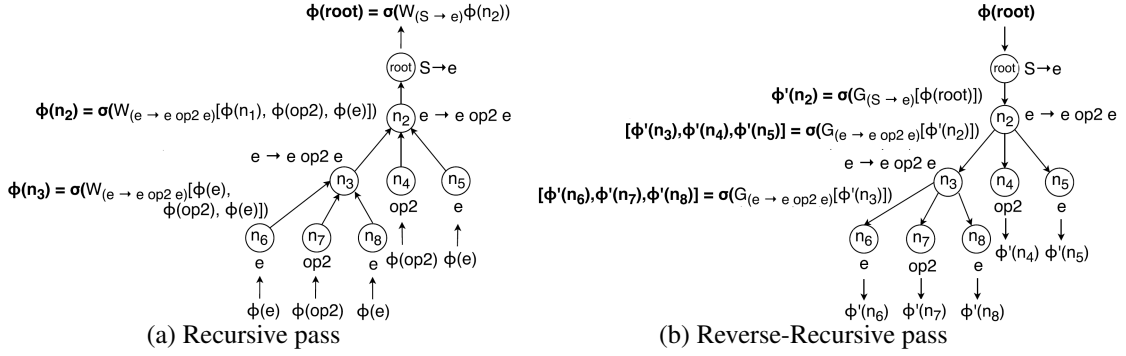

Figure 3: (a) The initial recursive pass of the R3NN. (b) The reverse-recursive pass of the R3NN where the input is the output of the previous recursive pass.

3. For every production rule $r \in R$, a deep neural network $f_r$ which takes as input a vector $x \in \mathbb{R}^{Q \cdot M}$, with $Q$ being the number of symbols on the RHS of the production rule $r$, and outputs a vector $y \in \mathbb{R}^M$. Therefore, the production-rule network $f_r$ takes as input a concatenation of the distributed representations of each of its RHS symbols and produces a distributed representation for the LHS symbol.

4. For every production rule $r \in R$, an additional deep neural network $g_r$ which takes as input a vector $x' \in \mathbb{R}^M$ and outputs a vector $y' \in \mathbb{R}^{Q \cdot M}$. We can think of $g_r$ as a reverse production-rule network that takes as input a vector representation of the LHS and produces a concatenation of the distributed representations of each of the rule's RHS symbols.

Let $E$ be the set of all valid expansions in a PPT $T$, let $L$ be the current leaf nodes of $T$ and $N$ be the current non-leaf (rule) nodes of $T$. Let $S(l)$ be the symbol of leaf $l \in L$ and $R(n)$ represent the production rule of non-leaf node $n \in N$.

### 4.1.1 GLOBAL TREE INFORMATION AT THE LEAVES

To compute the probability distribution over the set $E$, the R3NN first computes a distributed representation for each leaf node that contains global tree information. To accomplish this, for every leaf node $l \in L$ in the tree we retrieve its distributed representation $\phi(S(l))$ . We now do a standard recursive bottom-to-top, RHS→LHS pass on the network, by going up the tree and applying $f_{R(n)}$ for every non-leaf node $n \in N$ on its RHS node representations (see Figure 3(a)). These networks $f_{R(n)}$ produce a node representation which is input into the parent's rule network and so on until we reach the root node.

Once at the root node, we effectively have a fixed-dimensionality global tree representation $\phi(root)$ for the start symbol. The problem is that this representation has lost any notion of tree position. To solve this problem R3NN now does what is effectively a reverse-recursive pass which starts at the root node with $\phi(root)$ as input and moves towards the leaf nodes (see Figure 3(b)).

More concretely, we start with the root node representation $\phi(root)$ and use that as input into the rule network $g_{R(root)}$ where $R(root)$ is the production rule that is applied to the start symbol in $T$. This produces a representation $\phi'(c)$ for each RHS node $c$ of $R(root)$. If $c$ is a non-leaf node, we iteratively apply this procedure to $c$, $i.e.$, process $\phi'(c)$ using $g_{R(c)}$ to get representations $\phi'(cc)$ for every RHS node $cc$ of $R(c)$, etc. If $c$ is a leaf node, we now have a leaf representation $\phi'(c)$ which has an information path to $\phi(root)$ and thus to every other leaf node in the tree. Once the reverse-recursive process is complete, we now have a distributed representation $\phi'(l)$ for every leaf node $l$ which contains global tree information. While $\phi(l_1)$ and $\phi(l_2)$ could be equal for leaf nodes which have the same symbol type, $\phi'(l_1)$ and $\phi'(l_2)$ will not be equal even if they have the same symbol type because they are at different positions in the tree.

### 4.1.2 EXPANSION PROBABILITIES

Given the global leaf representations $\phi'(l)$, we can now straightforwardly acquire scores for each $e \in E$. For expansion $e$, let $e.r$ be the expansion type (production rule $r \in R$ that $e$ applies) and let $e.l$ be the leaf node $l$ that $e.r$ is applied to. $z_e = \phi'(e.l) \cdot \omega(e.r)$ The score of an expansion is calculated using $z_e = \phi'(e.l) \cdot \omega(e.r)$. The probability of expansion $e$ is simply the exponentiated normalized sum over all scores: $\pi(e) = \frac{e^{z_e}}{\sum_{e' \in E} e^{z_{e'}}}$.

An additional improvement that was found to help was to add a bidirectional LSTM (BLSTM) to process the global leaf representations right before calculating the scores. To do this, we first order the global leaf representations sequentially from left-most leaf node to right-mode leaf node. We then treat each leaf node as a time step for a BLSTM to process. This provides a sort of skip connection between leaf nodes, which potentially reduces the path length that information needs to travel between leaf nodes in the tree. The BLSTM hidden states are then used in the score calculation rather than the leaves themselves.

The R3NN can be seen as an extension and combination of several previous tree-based models, which were mainly developed in the context of natural language processing (Le & Zuidema, 2014; Paulus et al., 2014; Irsoy & Cardie, 2013).

## 5 CONDITIONING WITH INPUT/OUTPUT EXAMPLES

Now that we have defined a generation process over tree-structured programs, we need a way of conditioning this generation process on a set of input/output examples. The set of input/output examples provide a nearly complete specification for the desired output program, and so a good encoding of the examples is crucial to the success of our program generator. For the most part, this example encoding needs to be domain-specific, since different DSLs have different inputs (some may operate over integers, some over strings, *etc.*). Therefore, in our case, we use an encoding adapted to the input-output strings that our DSL operates over. We also investigate different ways of conditioning program search on the learnt example input-output encodings.

### 5.1 ENCODING INPUT/OUTPUT EXAMPLES

There are two types of information that string manipulation programs need to extract from input-output examples: 1) constant strings, such as "@domain.com" or ".", which appear in all output examples; 2) substring indices in input where the index might be further defined by a regular expression. These indices determine which parts of the input are also present in the output. To simplify the DSL, we assume that there is a fixed finite universe of possible constant strings that could appear in programs. Therefore we focus on extracting the second type of information, the substring indices.

In earlier hand-engineered systems such as FlashFill, this information was extracted from the input-output strings by running the Longest Common Substring algorithm, a dynamic programming algorithm that efficiently finds matching substrings in string pairs. To extract substrings, FlashFill runs LCS on every input-output string pair in the I/O set to get a set of substring candidates. It then takes the entire set of substring candidates and simply tries every possible regex and constant index that can be used at substring boundaries, exhaustively searching for the one which is the most "general", where generality is specified by hand-engineered heuristics.

In contrast to these previous methods, instead of hand-designing a complicated algorithm to extract regex-based substrings, we develop neural network based architectures that are capable of learning to extract and produce continuous representations of the likely regular expressions given I/O examples.

### 5.1.1 BASELINE LSTM ENCODER

Our first I/O encoding network involves running two separate deep bidirectional LSTM networks for processing the input and the output string in each example pair. For each pair, it then concatenates the topmost hidden representation at every time step to produce a $4HT$-dimensional feature vector per I/O pair, where $T$ is the maximum string length for any input or output string, and $H$ is the topmost LSTM hidden dimension.

We then concatenate the encoding vectors across all I/O pairs to get a vector representation of the entire I/O set. This encoding is conceptually straightforward and has very little prior knowledge about what operations are being performed over the strings, *i.e.*, substring, constant, etc., which might make it difficult to discover substring indices, especially the ones based on regular expressions.

### 5.1.2 CROSS CORRELATION ENCODER

To help the model discover input substrings that are copied to the output, we designed an novel I/O example encoder to compute the cross correlation between each input and output example representation. We used the two output tensors of the LSTM encoder (discussed above) as inputs to this encoder. For each example pair, we first slide the output feature block over the input feature block and compute the dot product between the respective position representation. Then, we sum over all overlapping time steps. Features of all pairs are then concatenated to form a $2 * (T-1)$-dimensional vector encoding for all example pairs. There are $2 * (T-1)$ possible alignments in total between input and output feature blocks. An illustration of the cross-correlation encoder is shown in Figure 9. We also designed the following variants of this encoder.

**Diffused Cross Correlation Encoder:** This encoder is identical to the Cross Correlation encoder except that instead of summing over overlapping time steps after the element-wise dot product, we simply concatenate the vectors corresponding to all time steps, resulting in a final representation that contains $2 * (T-1) * T$ features for each example pair.

**LSTM-Sum Cross Correlation Encoder:** In this variant of the Cross Correlation encoder, instead of doing an element-wise dot product, we run a bidirectional LSTM over the concatenated feature blocks of each alignment. We represent each alignment by the LSTM hidden representation of the final time step leading to a total of $2 * H * 2 * (T-1)$ features for each example pair.

**Augmented Diffused Cross Correlation Encoder:** For this encoder, the output of each character position of the Diffused Cross Correlation encoder is combined with the character embedding at this position, then a basic LSTM encoder is run over the combined features to extract a $4 * H$-dimensional vector for both the input and the output streams. The LSTM encoder output is then concatenated with the output of the Diffused Cross Correlation encoder forming a $(4 * H + T * (T-1))$-dimensional feature vector for each example pair.

### 5.2 CONDITIONING PROGRAM SEARCH ON EXAMPLE ENCODINGS

Once the I/O example encodings have been computed, we can use them to perform conditional generation of the program tree using the R3NN model. There are a number of ways in which the PPT generation model can be conditioned using the I/O example encodings depending on where the I/O example information is inserted in the R3NN model. We investigated three locations to inject example encodings:

**1) Pre-conditioning:** where example encodings are concatenated to the encoding of each tree leaf, and then passed to a conditioning network before the bottom-up recursive pass over the program tree. The conditioning network can be either a multi-layer feedforward network, or a bidirectional LSTM network running over tree leaves. Running an LSTM over tree leaves allows the model to learn more about the relative position of each leaf node in the tree.

**2) Post-conditioning:** After the reverse-recursive pass, example encodings are concatenated to the updated representation of each tree leaf and then fed to a conditioning network before computing the expansion scores.

**3) Root-conditioning:** After the recursive pass over the tree, the root encoding is concatenated to the example encodings and passed to a conditioning network. The updated root representation is then used to drive the reverse-recursive pass.

Empirically, pre-conditioning worked better than either root- or post- conditioning. In addition, conditioning at all 3 places simultaneously did not cause a significant improvement over just pre-conditioning. Therefore, for the experimental section, we report models which only use pre-conditioning.

# 6 EXPERIMENTS

In order to evaluate and compare variants of the previously described models, we generate a dataset randomly from the DSL. To do so, we first enumerate all possible programs under the DSL up to a specific number of instructions, which are then partitioned into training, validation and test sets. In order to have a tractable number of programs, we limited the maximum number of instructions for programs to be 13. Length 13 programs are important for this specific DSL because all larger programs can be written as compositions of sub-programs of length at most 13. The semantics of length 13 programs therefore constitute the "atoms" of this particular DSL.

In testing our model, there are two different categories of generalization. The first is input/output generalization, where we are given a new set of input/output examples as well as a program with a specific tree that we have seen during training. This represents the model's capacity to be applied on new data. The second category is program generalization, where we are given both a previously unseen program tree in addition to unseen input/output examples. Therefore the model needs to have a sufficient enough understanding of the semantics of the DSL that it can construct novel combinations of operations. For all reported results, training sets correspond to the first type of generalization since we have seen the program tree but not the input/output pairs. Test sets represent the second type of generalization, as they are trees which have not been seen before on input/output pairs that have also not been seen before.

In this section, we compare several different variants of our model. We first evaluate the effect of each of the previously described input/output encoders. We then evaluate the R3NN model against a simple recurrent model called io2seq, which is basically an LSTM that takes as input the input/output conditioning vector and outputs a sequence of DSL symbols that represents a linearized program tree. Finally, we report the results of the best model on the length 13 training and testing sets, as well as on a set of 238 benchmark functions.

## 6.1 SETUP AND HYPERPARAMETERS SETTINGS

For training the R3NN, two hyperparameters that were crucial for stabilizing training were the use of hyperbolic tangent activation functions in both R3NN (other activations such as ReLU more consistently diverged during our initial experiments) and cross-correlation I/O encoders and the use of minibatches of length 8. Additionally, for all results, the program tree generation is conditioned on a set of 10 input/output string pairs. We used ADAM (Kingma & Ba, 2014) to optimize the networks with a learning rate of 0.001. Network weights used the default torch initializations.

Due to the difficulty of batching tree-based neural networks since each sample in a batch has a potentially different tree structure, we needed to do batching sequentially. Therefore for each mini-batch of size $N$, we accumulated the gradients for each sample. After all N sample gradients were accumulated, we updated the parameters and reset the accumulated gradients. Due to this sequential processing, in order to train models in a reasonable time, we limited our batch sizes to between 8-12. Despite the computational inefficiency, batching was critical to successfully train an R3NN, as online learning often caused the network to diverge.

For each latent function and set of input/output examples that we test on, we report whether we had a success after sampling 100 functions from the model and testing all 100 to see if one of these functions is equivalent to the latent function. Here we consider two functions to be equivalent with respect to a specific input/output example set if the functions output the same strings when run on the inputs. Under this definition, two functions can have a different set of operations but still be equivalent with respect to a specific input-output set.

We restricted the maximum size of training programs to be 13 because of two computational considerations. As described earlier, one difficulty was in batching tree-based neural networks of different structure and the computational cost of batching increases with the increase in size of the program trees. The second issue is that valid I/O strings for programs often grow with the program length, in the sense that for programs of length 40 a minimal valid I/O string will typically be much longer than a minimal valid I/O string for length 20 programs. For example, for a program such as (Concat (ConstStr "longstring") (Concat (ConstStr "longstring") (Concat (ConstStr "longstring") ...))), the valid output string would be "longstringlongstringlongstring..." which could be many

| I/O Encoding | Train | Test |
|---|---|---|
| LSTM | 88% | 88% |
| Cross Correlation (CC) | 67% | 65% |
| Diffused CC | 89% | 88% |
| LSTM-sum CC | 90% | 91% |
| Augmented diffused CC | 91% | 91% |

Table 1: The effect of different input/output encoders on accuracy. Each result used 100 samples. There is almost no generalization error in the results.

| Sampling | Train | Test |
|---|---|---|
| io2seq | 44% | 42% |

Table 2: Testing the I/O-vector-to-sequence model. Each result used 100 samples.

hundreds of characters long. Because of limited GPU memory, the I/O encoder models can quickly run out of memory.

## 6.2 EXAMPLE ENCODING

In this section, we evaluate the effect of several different input/output example encoders. To control for the effect of the tree model, all results here used an R3NN with fixed hyperparameters to generate the program tree. Table 1 shows the performance of several of these input/output example encoders. We can see that the summed cross-correlation encoder did not perform well, which can be due to the fact that the sum destroys positional information that might be useful for determining specific substring indices. The LSTM-sum and the augmented diffused cross-correlation models did the best. Surprisingly, the LSTM encoder was capable of finding nearly 88% of all programs without having any prior knowledge explicitly built into the architecture. We use 100 samples for evaluating the Train and Test sets. The training performance is sometimes slightly lower because there are close to 5 million training programs but we only look at less than 2 million of these programs during training. We sample a subset of only 1000 training programs from the 5 million program set to report the training results in the tables. The test sets also consist of 1000 programs.

## 6.3 IO2SEQ

In this section, we motivate the use of the R3NN by testing whether a simpler model can also be used to generate programs. The io2seq model is an LSTM whose initial hidden and cell states are a function of the input/output encoding vector. The io2seq model then generates a linearized tree of a program symbol-by-symbol. An example of what a linearized program tree looks like is $(_S(_e(_f(_{ConstStr}\text{``@''})_{ConstStr})_f)_e)_S$, which represents the program tree that returns the constant string "@". Predicting a linearized tree using an LSTM was also done in the context of parsing (Vinyals et al., 2015). For the io2seq model, we used the LSTM-sum cross-correlation I/O conditioning model.

The results in Table 2 show that the performance of the io2seq model at 100 samples per latent test function is far worse than the R3NN, at around 42% versus 91%, respectively. The reasons for that could be that the io2seq model needs to perform far more decisions than the R3NN, since the io2seq model has to predict the parentheses symbols that determine at which level of the tree a particular symbol is at. For example, the io2seq model requires on the order of 100 decisions for length 13 programs, while the R3NN requires no more than 13.

## 6.4 EFFECT OF SAMPLING MULTIPLE PROGRAMS

For the best R3NN model that we trained, we also evaluated the effect that a different number of samples per latent function had on performance. The results are shown in Table 3. The increase of the model's performance as the sample size increases hints that the model has a notion of what type of program satisfies a given I/O pair, but it might not be that certain about the details such as which regex to use, etc. By 300 samples, the model is nearing perfect accuracy on the test sets.

| Sampling | Train | Test |
|---|---|---|
| 1-best | 60% | 63% |
| 1-sample | 56% | 57% |
| 10-sample | 81% | 79% |
| 50-sample | 91% | 89% |
| 100-sample | 94% | 94% |
| 300-sample | 97% | 97% |

Table 3: The effect of sampling multiple programs on accuracy. 1-best is deterministically choosing the expansion with highest probability at each step.

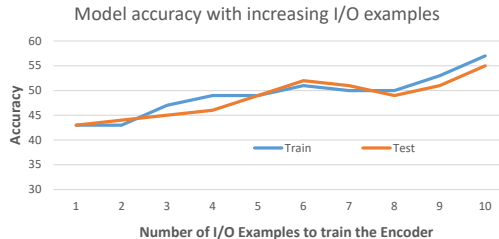

Figure 4: The train and test accuracies for models trained with different number of input-output examples.

## 6.5    EFFECT OF NUMBER OF INPUT-OUTPUT EXAMPLES

We evaluate the effect of varying the number of input-output examples used to train the Input-output encoders. The 1-best accuracy for train and test data for models trained for 74 epochs is shown in Figure 4. As expected, the accuracy increases with increase in number of input-output examples, since more examples add more information to the encoder and constrain the space of consistent programs in the DSL.

## 6.6    FLASHFILL BENCHMARKS

We also evaluate our learnt models on 238 real-world FlashFill benchmarks obtained from the Microsoft Excel team and online help-forums. These benchmarks involve string manipulation tasks described using input-output examples. We evaluate two models – one with a cross correlation encoder trained on 5 input-output examples and another trained on 10 input-output examples. Both the models were trained on randomly sampled programs from the DSL upto size 13 with randomly generated input-output examples.

The distribution of the size of smallest DSL programs needed to solve the benchmark tasks is shown in Figure 5(a), which varies from 4 to 63. The figure also shows the number of benchmarks for which our model was able to learn the program using 5 input-output examples using samples of top-2000 learnt programs. In total, the model is able to learn programs for 91 tasks (38.2%). Since the model was trained for programs upto size 13, it is not surprising that it is not able to solve tasks that need larger program size. There are 110 FlashFill benchmarks that require programs upto size 13, out of which the model is able to solve 82.7% of them.

The effect of sampling multiple learnt programs instead of only top program is shown in Figure 5(b). With only 10 samples, the model can already learn about 13% of the benchmarks. We observe a steady increase in performance upto about 2000 samples, after which we do not observe any significant improvement. Since there are more than 2 million programs in the DSL of length 11 itself, the enumerative techniques with uniform search do not scale well (Alur et al., 2015).

We also evaluate a model that is learnt with 10 input-output examples per benchmark. This model can only learn programs for about 29% of the FlashFill benchmarks. Since the FlashFill benchmarks contained only 5 input-output examples for each task, to run the model that took 10 examples as input, we duplicated the I/O examples. Our models are trained on the synthetic training dataset

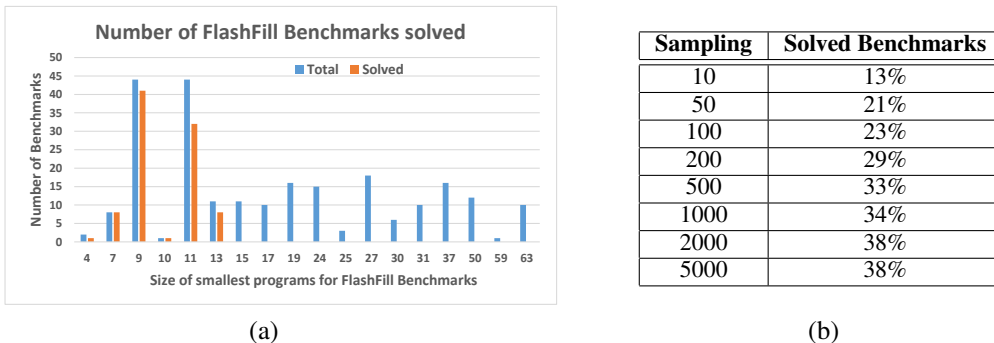

| Sampling | Solved Benchmarks |
| --- | --- |
| 10 | 13% |
| 50 | 21% |
| 100 | 23% |
| 200 | 29% |
| 500 | 33% |
| 1000 | 34% |
| 2000 | 38% |
| 5000 | 38% |

(a)                                                                          (b)

Figure 5: (a) The distribution of size of programs needed to solve FlashFill tasks and the performance of our model, (b) The effect of sampling for trying top-k learnt programs.

| Input $v$ | Output |
| --- | --- |
| [CPT-00350 | [CPT-00350] |
| [CPT-00340] | [CPT-00340] |
| [CPT-114563] | [CPT-114563] |
| [CPT-1AB02] | [CPT-1AB02] |
| [CPT-00360 | [CPT-00360] |

(a)

| Input $v$ | Output |
| --- | --- |
| 732606129 | 0x73 |
| 430257526 | 0x43 |
| 444004480 | 0x44 |
| 371255254 | 0x37 |
| 635272676 | 0x63 |

(b)

| Input $v$ | Output |
| --- | --- |
| John Doyle | John D. |
| Matt Walters | Matt W. |
| Jody Foster | Jody F. |
| Angela Lindsay | Angela L. |
| Maria Schulte | Maria S. |

(c)

Figure 6: Some example solved benchmarks: (a) cleaning up medical codes with closing brackets, (b) generating Hex numbers with first two digits, (c) transforming names to firstname and last initial.

that is generated uniformly from the DSL. Because of the discrepancy between the training data distribution (uniform) and auxiliary task data distribution, the model with 10 input/output examples might not perform the best on the FlashFill benchmark distribution, even though it performs better on the synthetic data distribution (on which it is trained) as shown in Figure 4.

Our model is able to solve majority of FlashFill benchmarks that require learning programs with upto 3 Concat operations. We now describe a few of these benchmarks, also shown in Figure 6. An Excel user wanted to clean a set of medical billing records by adding a missing "]" to medical codes as shown in Figure 6(a). Our system learns the following program given these 5 input-output examples: Concat(SubStr($v$,ConstPos(0),($d$,-1,End)), ConstStr("]")). The program concatenates the substring between the start of the input string and the position of the last digit regular expression with the constant string "]". Another task that required user to transform some numbers into a hex format is shown in Figure 6(b). Our system learns the following program: Concat(ConstStr("0x"),SubStr($v$,ConstPos(0),ConstPos(2))). For some benchmarks with long input strings, it is still able to learn regular expressions to extract the desired substring, e.g. it learns a program to extract "NancyF" from the string "123456789,freehafer ,drew ,nancy,19700101,11/1/2007,NancyF@north.com,1230102,123 1st Avenue,Seattle,wa,09999".

Our system is currently not able to learn programs for benchmarks that require 4 or more Concat operations. Two such benchmarks are shown in Figure 7. The task of combining names in Figure 7(a) requires 6 Concat arguments, whereas the phone number transformation task in Figure 7(b) requires 5 Concat arguments. This is mainly because of the scalability issues in training with programs of larger size. There are also a few interesting benchmarks where the R3NN models gets very close to learning the desired program. For example, for the task "Bill Gates" → "Mr. Bill Gates", it learns a program that generates "Mr.Bill Gates" (without the whitespace), and for the task "617-444-5454" → "(617) 444-5454", it learns a program that generates the string "(617 444-5454".

| | Input $v$ | Output |
|---|---|---|
| 1 | John James Paul | John, James, and Paul. |
| 2 | Tom Mike Bill | Tom, Mike, and Bill. |
| 3 | Marie Nina John | Marie, Nina, and John. |
| 4 | Reggie Anna Adam | Reggie, Anna, and Adam. |

(a)

| | Input $v$ | Output |
|---|---|---|
| 1 | (425) 221 6767 | 425-221-6767 |
| 2 | 206.225.1298 | 206-225-1298 |
| 3 | 617-224-9874 | 617-224-9874 |
| 4 | 425.118.9281 | 425-118-9281 |

(b)

Figure 7: Some unsolved benchmarks: (a)Combining names by different delimiters. (b) Transforming phone numbers to consistent format.

## 7 RELATED WORK

We have seen a renewed interest in recent years in the area of Program Induction and Synthesis.

In the machine learning community, a number of promising neural architectures have been proposed to perform *program induction*. These methods have employed architectures inspired from computation modules (Turing Machines, RAM) (Graves et al., 2014; Kurach et al., 2015; Reed & de Freitas, 2015; Neelakantan et al., 2015) or common data structures such as stacks used in many algorithms (Joulin & Mikolov, 2015). These approaches represent the atomic operations of the network in a differentiable form, which allows for efficient end-to-end training of a neural controller. However, unlike our approach that learns comprehensible complete programs, many of these approaches learn only the program behavior (*i.e.*, they produce desired outputs on new input data). Some recently proposed methods (Kurach et al., 2015; Gaunt et al., 2016; Riedel et al., 2016; Bunel et al., 2016) do learn interpretable programs but these techniques require learning a separate neural network model for each individual task, which is undesirable in many synthesis settings where we would like to learn programs in real-time for a large number of tasks. Liang et al. (2010) restrict the problem space with a probabilistic context-free grammar and introduce a new representation of programs based on combinatory logic, which allows for sharing sub-programs across multiple tasks. They then take a hierarchical Bayesian approach to learn frequently occurring substructures of programs. Our approach, instead, uses neural architectures to condition the search space of programs, and does not require additional step of representing program space using combinatory logic for allowing sharing.

The DSL-based program synthesis approach has also seen a renewed interest recently (Alur et al., 2015). It has been used for many applications including synthesizing low-level bitvector implementations (Solar-Lezama et al., 2005), Excel macros for data manipulation (Gulwani, 2011; Gulwani et al., 2012), superoptimization by finding smaller equivalent loop bodies (Schkufza et al., 2013), protocol synthesis from scenarios (Udupa et al., 2013), synthesis of loop-free programs (Gulwani et al., 2011), and automated feedback generation for programming assignments (Singh et al., 2013). The synthesis techniques proposed in the literature generally employ various search techniques including enumeration with pruning, symbolic constraint solving, and stochastic search, while supporting different forms of specifications including input-output examples, partial programs, program invariants, and reference implementation.

In this paper, we consider input-output example based specification over the hypothesis space defined by a DSL of string transformations, similar to that of FlashFill (without conditionals) (Gulwani, 2011). The key difference between our approach over previous techniques is that our system is trained completely in an end-to-end fashion, while previous techniques require significant manual effort to design heuristics for efficient search. There is some work on guiding the program search using learnt clues that suggest likely DSL expansions, but the clues are learnt over hand-coded textual features of examples (Menon et al., 2013). Moreover, their DSL consists of composition of about 100 high-level text transformation functions such as count and dedup, whereas our DSL consists of tree structured programs over richer regular expression based substring constructs.

There is also a recent line of work on learning probabilistic models of code from a large number of code repositories (*big code*) (Raychev et al., 2015; Bielik et al., 2016; Hindle et al., 2016), which are then used for applications such as auto-completion of partial programs, inference of variable and method names, program repair, etc. These language models typically capture only the syntactic

properties of code, unlike our approach that also tries to capture the semantics to learn the desired program. The work by Maddison & Tarlow (2014) addresses the problem of learning structured generative models of source code but both their model and application domain are different from ours. Piech et al. (2015) use an NPM-RNN model to embed program ASTs, where a subtree of the AST rooted at a node n is represented by a matrix obtained by combining representations of the children of node n and the embedding matrix of the node n itself (which corresponds to its functional behavior). The forward pass in our R3NN architecture from leaf nodes to the root node is, at a high-level, similar, but we use a distributed representation for each grammar symbol that leads to a different root representation. Moreover, R3NN also performs a reverse-recursive pass to ensure all nodes in the tree encode global information about other nodes in the tree. Finally, the R3NN network is then used to incrementally build a tree to synthesize a program.

The R3NN model employed in our work is related to several tree and graph structured neural networks present in the NLP literature (Le & Zuidema, 2014; Paulus et al., 2014; Irsoy & Cardie, 2013). The Inside-Outside Recursive Neural Network (Le & Zuidema, 2014) in particular is most similar to the R3NN, where they generate a parse tree incrementally by using global leaf-level representations to determine which expansions in the parse tree to take next.

# 8 CONCLUSION

We have proposed a novel technique called *Neuro-Symbolic Program Synthesis* that is able to construct a program incrementally based on given input-output examples. To do so, a new neural architecture called Recursive-Reverse-Recursive Neural Network is used to encode and expand a partial program tree into a full program tree. Its effectiveness at example-based program synthesis is demonstrated, even when the program has not been seen during training.

These promising results open up a number of interesting directions for future research. For example, we took a supervised-learning approach here, assuming availability of target programs during training. In some scenarios, we may only have access to an oracle that returns the desired output given an input. In this case, reinforcement learning is a promising framework for program synthesis.

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

$$
\begin{aligned}
[\![\mathrm{Concat}(f_1, \cdots, f_n)]\!]_v &= \mathrm{Concat}([\![f_1]\!]_v, \cdots, [\![f_n]\!]_v) \\
[\![\mathrm{ConstStr}(s)]\!]_v &= s \\
[\![\mathrm{SubStr}(v, p_l, p_r)]\!]_v &= v[[\![p_l]\!]_v .. [\![p_r]\!]_v] \\
[\![\mathrm{ConstPos}(k)]\!]_v &= k > 0?\ k : \mathrm{len}(s) + k \\
[\![(r, k, \mathrm{Start})]\!]_v &= \text{Start of } k^{\mathrm{th}}\text{match of r in v} \\
&\quad \text{from beginning (end if } k < 0) \\
[\![(r, k, \mathrm{End})]\!]_v &= \text{End of } k^{\mathrm{th}}\text{match of r in v} \\
&\quad \text{from beginning (end if } k < 0)
\end{aligned}
$$

Figure 8: The semantics of the DSL for string transformations.

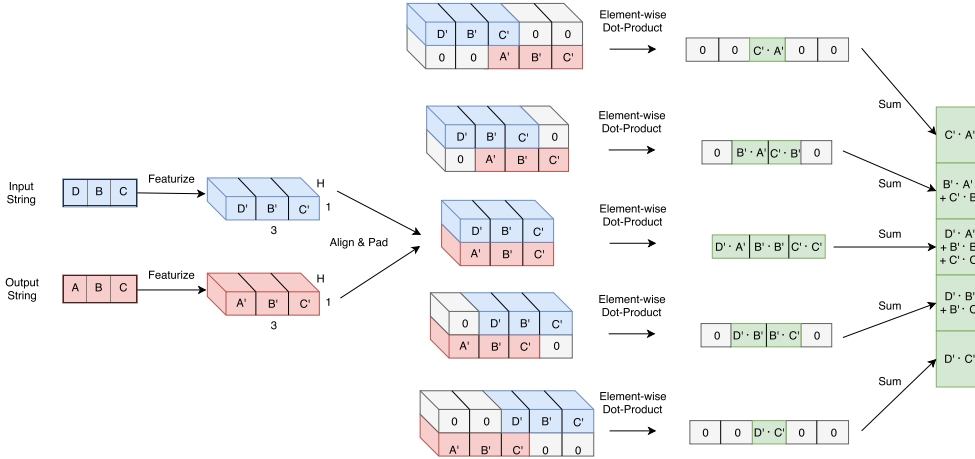

Figure 9: The cross correlation encoder to encode a single input-output example.

Vinyals, Oriol, Kaiser, Lukasz, Koo, Terry, Petrov, Slav, Sutskever, Ilya, and Hinton, Geoffrey. Grammar as a foreign language. In *ICLR*, 2015.

## A DOMAIN-SPECIFIC LANGUAGE FOR STRING TRANSFORMATIONS

The semantics of the DSL programs is shown in Figure 8. The semantics of a Concat expression is to concatenate the results of recursively evaluating the constituent substring expressions $f_i$. The semantics of ConstStr(s) is to simply return the constant string $s$. The semantics of a substring expression is to first evaluate the two position logics $p_l$ and $p_r$ to $p_1$ and $p_2$ respectively, and then return the substring corresponding to $v[p_1..p_2]$. We denote $s[i..j]$ to denote the substring of string $s$ starting at index i (inclusive) and ending at index j (exclusive), and len(s) denotes its length. The semantics of ConstPos(k) expression is to return $k$ if $k > 0$ or return len $+ k$ (if $k < 0$). The semantics of position logic $(r, k, \mathrm{Start})$ is to return the Start of $k^{\mathrm{th}}$ match of r in $v$ from the beginning (if $k > 0$) or from the end (if $k < 0$).

