# Peer review of "Neuro-Symbolic Program Synthesis"

_ICLR 2017 — accepted_

[Public Comment · (anonymous) · 15 Dec 2016]
**Training?**

Few remarks:

The paper explains what the tree is and how examples are encoded, but its missing important explanation on:

- how the I/O examples are encoded in the tree during training exactly.
- it is not well explained how the prediction works during testing when the examples are given.

It looks like the DSL is restricted to know all constant strings that will be used, which seems difficult in realistic scenarios.

[Official Review · AnonReviewer2 · rating 8 · confidence 4 · 16 Dec 2016]
**Nice program synthesis approach to a practical Excel flash-fill like application**

This paper proposes a model that is able to infer a program from input/output example pairs, focusing on a restricted domain-specific language that captures a fairly wide variety of string transformations, similar to that used by Flash Fill in Excel.  The approach is to model successive “extensions” of a program tree conditioned on some embedding of the input/output pairs.  Extension probabilities are computed as a function of leaf and production rule embeddings — one of the main contributions is the so-called “Recursive-Reverse-Recursive Neural Net” which computes a globally aware embedding of a leaf by doing something that looks like belief propagation on a tree (but training this operation in an end-to-end differentiable way).

There are many strong points about this paper.  In contrast with some of the related work in the deep learning community, I can imagine this being used in an actual application in the near future.  The R3NN idea is a good one and the authors motivate it quite well.  Moreover, the authors have explored many variants of this model to understand what works well and what does not.  Finally, the exposition is clear (even if it is a long paper), which made this paper a pleasure to read.  Some weaknesses of this paper: the results are still not super accurate, perhaps because the model has only been trained on small programs but is being asked to infer programs that should be much longer.  And it’s unclear why the authors did not simply train on longer programs…  It also seems that the number of I/O pairs is fixed?  So if I had more I/O pairs, the model might not be able to use those additional pairs (and based on the experiments, more pairs can hurt…).  Overall however, I would certainly like to see this paper accepted at ICLR.

Other miscellaneous comments:
* Too many e’s in the expansion probability expression — might be better just to write “Softmax”.
* There is a comment about adding a bidirectional LSTM to process the global leaf representations before calculating scores, but no details are given on how this is done (as far as I can see).
* The authors claim that using hyperbolic tangent activation functions is important — I’d be interested in some more discussion on this and why something like ReLU would not be good.
* It’s unclear to me how batching was done in this setting since each program has a different tree topology.  More discussion on this would be appreciated.  Related to this, it would be good to add details on optimization algorithm (SGD?  Adagrad?  Adam?), learning rate schedules and how weights were initialized.  At the moment, the results are not particularly reproducible.
* In Figure 6 (unsolved benchmarks), it would be great to add the program sizes for these harder examples (i.e., did the approach fail because these benchmarks require long programs?  Or was it some other reason?)
* There is a missing related work by Piech et al (Learning Program Embeddings…) where the authors trained a recursive neural network (that matched abstract syntax trees for programs submitted to an online course) to predict program output (but did not synthesize programs).

[Official Review · AnonReviewer1 · rating 7 · confidence 3 · 20 Dec 2016]
**Strong ideas for an important problem**

This paper sets out to tackle the program synthesis problem: given a set of input/output pairs discover the program that generated them. The authors propose a bipartite model, with one component that is a generative model of tree-structured programs and the other component an input/output pair encoder for conditioning. They consider applying many variants of this basic model to a FlashFill DSL. The experiments explore a practical dataset and achieve fine numbers. The range of models considered, carefulness of the exposition, and basic experimental setup make this a valuable paper for an important area of research. I have a few questions, which I think would strengthen the paper, but think it's worth accepting as is.

Questions/Comments:

- The dataset is a good choice, because it is simple and easy to understand. What is the effect of the "rule based strategy" for computing well formed input strings?

- Clarify what "backtracking search" is? I assume it is the same as trying to generate the latent function? 

- In general describing the accuracy as you increase the sample size could be summarize simply by reporting the log-probability of the latent function. Perhaps it's worth reporting that? Not sure if I missed something.

[Official Review · AnonReviewer3 · rating 5 · confidence 4 · 02 Jan 2017]

The paper presents a method to synthesize string manipulation programs based on a set of input output pairs. The paper focuses on a restricted class of programs based on a simple context free grammar sufficient to solve string manipulation tasks from the FlashFill benchmark. A probabilistic generative model called Recursive-Reverse-Recursive Neural Network (R3NN) is presented that assigns a probability to each program's parse tree after a bottom-up and a top-down pass. Results are presented on a synthetic dataset and a Microsoft Excel benchmark called FlashFill.

The problem of program synthesis is important with a lot of recent interest from the deep learning community. The approach taken in the paper based on parse trees and recursive neural networks seems interesting and promising. However, the model seems too complicated and unclear at several places (details below). On the negative side, the experiments are particularly weak, and the paper does not seem ready for publication based on its experimental results. I was positive about the paper until I realized that the method obtains an accuracy of 38% on FlashFill benchmark when presented with only 5 input-output examples but the performance degrades to 29% when 10 input-output examples are used. This was surprising to the authors too, and they came up with some hypothesis to explain this phenomenon. To me, this is a big problem indicating either a bug in the code or a severe shortcoming of the model. Any model useful for program synthesis needs to be applicable to many input-output examples because most complicated programs require many examples to disambiguate the details of the program.

Given the shortcoming of the experiments, I am not convinced that the paper is ready for publication. Thus, I recommend weak reject. I encourage the authors to address the comments below and resubmit as the general idea seems promising.

More comments:

I am unclear about the model at several places:
- How is the probability distribution normalized? Given the nature of bottom-up top-down evaluation of the potentials, should one enumerate over different completions of a program and the compare their exponentiated potentials? If so, does this restrict the applicability of the model to long programs as the enumeration of the completions gets prohibitively slow?
- What if you only use 1 input-output pair for each program instead of 5? Do the results get better?
- Section 5.1.2 is not clear to me. Can you elaborate by potentially including some examples? Does your input-output representation pre-supposes a fixed number of input-output examples across tasks (e.g. 5 or 10 for all of the tasks)?

Regarding the experiments,
- Could you present some baseline results on FlashFill benchmark based on previous work?
- Is your method only applicable to short programs? (based on the choice of 13 for the number of instructions)
- Does a program considered correct when it is identical to a test program, or is it considered correct when it succeeds on a set of held-out input-output pairs?
- When using 100 or more program samples, do you report the accuracy of the best program out of 100 (i.e. recall) or do you first filter the programs based on training input-output pairs and then evaluate a program that is selected?

Your paper is well beyond the recommended limit of 8 pages. please consider making it shorter.

[Public Comment · ICLR 2017 conference · 16 Jan 2017]
**Reactions to author response?**

Dear reviewers, do you have any reactions after the authors responded to your reviews?

[Final Decision · Program Chairs · 06 Feb 2017]
**ICLR committee final decision**

There is a bit of a spread in the reviewer scores and unfortunately it wasn't possible to entice the reviewers to participate in a discussion. The area chair therefore discounts the late review of reviewer3, who seems to have had a misunderstanding that was successfully rebutted by the authors. The other reviewers are supportive of the paper.